# Dispersal-induced instability in complex ecosystems

Joseph W. Baron ⬡ [1,2]✉ & Tobias Galla ⬡ [1,2]

In his seminal work in the 1970s, Robert May suggested that there is an upper limit to the number of species that can be sustained in stable equilibrium by an ecosystem. This deduction was at odds with both intuition and the observed complexity of many natural ecosystems. The so-called stability-diversity debate ensued, and the discussion about the factors contributing to ecosystem stability or instability continues to this day. We show in this work that dispersal can be a destabilising influence. To do this, we combine ideas from Alan Turing's work on pattern formation with May's random-matrix approach. We demonstrate how a stable equilibrium in a complex ecosystem with trophic structure can become unstable with the introduction of dispersal in space, and we discuss the factors which contribute to this effect. Our work highlights that adding more details to the model of May can give rise to more ways for an ecosystem to become unstable. Making May's simple model more realistic is therefore unlikely to entirely remove the upper bound on complexity.

[1] Department of Physics and Astronomy, School of Natural Sciences, The University of Manchester, Manchester M13 9PL, UK. [2] Instituto de Física Interdisciplinar y Sistemas Complejos IFISC (CSIC-UIB), 07122 Palma de Mallorca, Spain. ✉email: josephbaron@ifisc.uib-csic.es

Providing a firm counterpoint to the view that greater eco-system complexity promoted stability[1–6], Robert May used a simple statistical model[7,8] to argue that increasing the number of species in an ecosystem could in fact reduce stability. By analysing the eigenvalues of a randomly constructed community matrix, May deduced the following criterion for stability[7]

$$\sigma^2 NC < 1. \tag{1}$$

In this criterion $\sigma^2$ is the variance in the inter-species interactions, $N$ is the number of species and $C$ is the connectance (the probability that any given pair of species interact with one another). May's result shows that stability is decided by the product $c = \sigma^2 NC$. We will call this quantity the 'complexity' of the ecosystem.

May's model suggests that more complex ecosystems tend toward instability. For a fixed variance of interactions, there is an upper bound to the number of species and food web connections that the ecosystem can sustain. This idea quickly became controversial and May's work sparked the so-called complexity-stability (or diversity-stability) debate, which continues to this day[5,6,9].

Since the 1970s, the discussion around stability has been made more precise and subtle. It is now understood that there are a number of senses in which an ecosystem can be unstable and, indeed, there are a number of ways one can define diversity[5,10]. An ecosystem can be unstable with respect to, for example, the introduction of new species, the extinction of existing species, or environmental changes. May's work specifically addresses stability with respect to fluctuations in species abundance.

In order to understand the influence of the many aspects of real ecosystems on stability, May's fairly austere model has since been augmented and improved upon. Features not captured by May's initial model include food-web structure (e.g., trophic levels, modularity and nestedness)[8,11–15], the feasibility of the equilibrium[16,17], nonlinearities and alternative interpretations of 'interaction strength'[18–20] and variability of the environment and of species' susceptibility to environmental change[21–24]. In many models of complex ecosystems, however, only the total abundance of each species is considered without appreciating how the members of that species are distributed in space[7,8,11–13,16,17,19–24]. In such models, there is no notion of space and hence no dispersal. In this work, we explicitly include the effects of diffusive dispersal in space and study the effects on stability.

Dispersal may intuitively be expected to be a homogenising and stabilising influence. As demonstrated recently in ref. [25], it can indeed stabilise equilibria in spatially heterogeneous ecosystems. Perhaps counter-intuitively, the insight of Turing's seminal work[26] was that dispersal can also destabilise a dynamical system. Such instability has been studied in meta-population[27,28] predator–prey models with small numbers of species[29–31] and numerically for food webs on networks[32]. We combine Turing's idea with May's random-matrix approach to show that a similar destabilising effect can be seen in models of complex ecosystems.

In order not to obscure the key effects at work, we opt to modify May's paradigmatic model sparingly. This allows us to highlight the consequences of the inclusion of dispersal. We suppose that the abundances of the species rest in a steady, homogeneous equilibrium. In order to study stability, we examine the Jacobian matrix governing perturbations in species abundance about this equilibrium. Like May, we ask what statistical properties are required of the Jacobian matrix in order for the ecosystem to return to equilibrium when perturbed.

Unlike May, however, we allow for trophic structure in our model. It is the combination of dispersal and trophic structure which gives rise to the Turing-type instability. For the sake of mathematical simplicity, we confine our model to only two broad trophic classifications: predator and prey species. The two groups of species are distinguished by statistical differences in their interactions. Our approach can be generalised to more complicated food web structures.

We now postulate the form of the Jacobian matrix central to our problem, $\mathbf{M}_q$. The elements of this matrix describe how spatial disturbances of wavelength $\lambda = 2\pi/q$ in the abundance of one species affect the abundances of the other species; $q$ is known as the wavenumber[33].

Because of the trophic structure of the community, the matrix $\mathbf{M}_q$ has a block structure where each block has different statistics. Similarly structured random matrices have been used in previous literature[34,35]. The matrix $\mathbf{M}_q$ is comprised of three terms: a diffusion term, and an intra-species interaction term and an inter-species interaction term. We write

$$\mathbf{M}_q = -q^2 \mathbf{D} - \mathbf{d} + \mathbf{A}. \tag{2}$$

The diffusion coefficients for prey and predator species are $D_u$ and $D_v$ respectively (Fig. 1). The interaction matrix $\mathbf{A}$ is modelled as having elements drawn from a correlated Gaussian ensemble, although other distributions may be used to obtain the same results (see Section S7 in the Supplementary Information). Further details on the structure of $\mathbf{M}_q$ and how one arrives at this form are given in Fig. 1 and in the Methods section.

The problem of analysing stability reduces to finding the eigenvalue spectrum of the matrix $\mathbf{M}_q$. If the eigenvalues of this

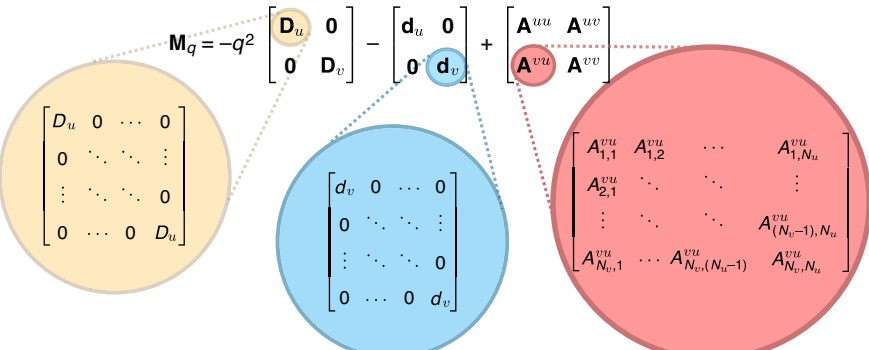

**Fig. 1 The structure of the stability matrix $\mathbf{M}_q$.** The matrix is composed of three parts: a diffusion matrix $\mathbf{D}$, a self-interaction matrix $\mathbf{d}$, and an interaction matrix $\mathbf{A}$ with entries drawn at random from a probability distribution. Each matrix is split into blocks due to the trophic structure of the community—we use the subscript $u$ to denote species which have mostly prey-like interactions and $v$ for species with mostly predator-like interactions. The approach can be generalised to more complicated block structures. The stability of the non-spatial ecosystem is described by the matrix for $q = 0$, or equivalently by setting $D_u = D_v = 0$ (see Methods and Section S1 in the Supplementary Information).

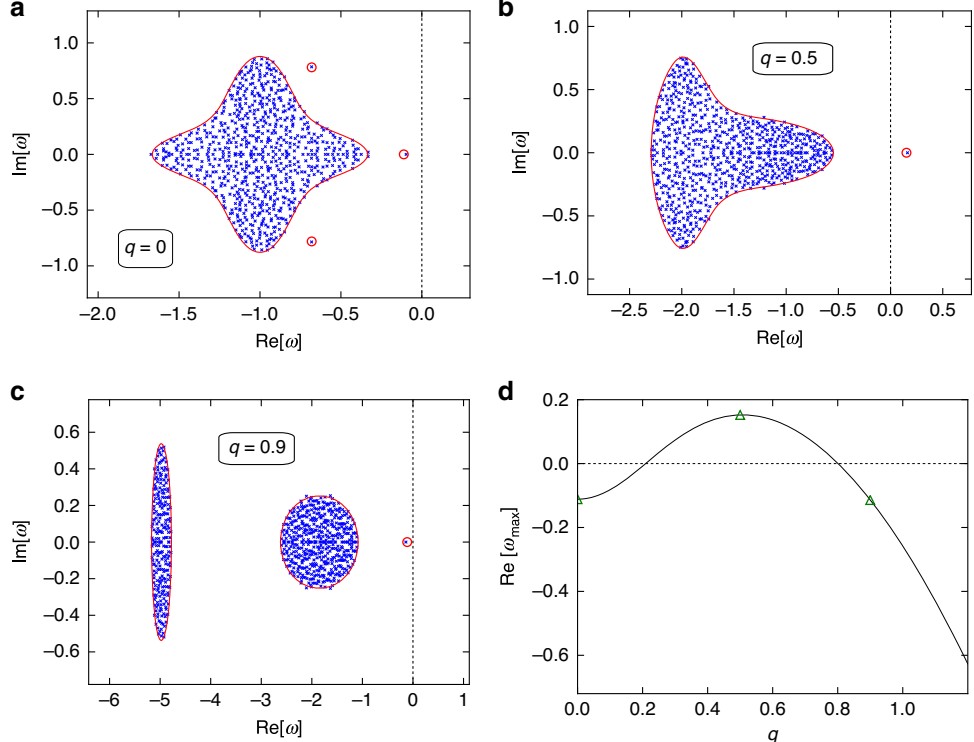

**Fig. 2 Eigenvalue spectra of the stability matrices in Fig. 1.** In May's original model, the eigenvalues all lay uniformly within a circle in the complex plane. In our model, the circle is warped and split into more complicated shapes. Also, a small number of outlier eigenvalues now stray from the bulk. Eigenvalues of computer-generated random matrices $\mathbf{M}_q$ are shown as blue crosses. They are compared to theoretical predictions for the boundary of the bulk region (red solid line) and theoretical predictions for the outliers (open red circles). For $q = 0$, all of the eigenvalues have negative real part (panel **a**)—the equilibrium is stable in the non-spatial ecosystem. For disturbances with larger wavenumber $q$, one of the eigenvalues crosses the real axis (panel **b**). The equilibrium is unstable against such perturbations. If the wavenumber $q$ is larger still, the rightmost eigenvalue returns to the negative half-plane (panel **c**). This is characteristic of a Turing instability[33]—the equilibrium is unstable with respect to disturbances of a finite range of wavelengths. This is shown in panel **d**, where we plot the real part of the rightmost eigenvalue as a function of $q$. The equilibrium is unstable against perturbations of wavenumber $q$ whenever $\mathrm{Re}\,[\omega_{max}] > 0$. The green triangles mark the wavenumbers from panels (**a**–**c**).

matrix all have negative real parts, then the equilibrium is stable with respect to disturbances of wavenumber $q$. Else, it is unstable. In order for the equilibrium to be stable on the whole, all eigenvalues of $\mathbf{M}_q$ must have negative real parts for all values of $q$.

If the number of species in the ecosystem is large, then the eigenvalue spectrum is dependent only on the statistics of $\mathbf{M}_q$ and not on its specific entries. Using random-matrix theory and ideas from statistical physics, we are able to deduce a mathematical expression for the support of the eigenvalue spectrum of $\mathbf{M}_q$. That is, we can find the region in the complex plane in which the eigenvalues sit and, most importantly, whether or not they have positive real parts. Examples are shown in Fig. 2.

With this analytical approach, we can calculate what properties of $\mathbf{M}_q$ make the equilibrium unstable. Thus, we can deduce how May's upper bound on ecosystem complexity is modified by the inclusion of dispersal and trophic structure. Most crucially, we show that equilibria which would be stable without spatial effects can be destabilised by dispersal. We find that this dispersal-induced instability is possible not only in a linear model but also in a non-linear system where the equilibrium is arrived at dynamically and hence is feasible by construction.

## Results

**Eigenvalue spectra.** We show some example eigenvalue spectra of the matrix $\mathbf{M}_q$ in Fig. 2. The vast majority of the eigenvalues group into a 'bulk' region, with the exception of a few outliers. These outliers cannot be ignored – the excursion of even one

eigenvalue across the imaginary axis to the positive real side makes the equilibrium unstable.

Using the statistical properties of $\mathbf{M}_q$ we are able to calculate mathematically the bulk regions to which most of the eigenvalues are confined and the locations of any outliers. In Fig. 2a–c, we show that these calculations agree very well with the spectra of computer-generated random matrices. We can therefore predict what community properties lead to stability or instability.

As Fig. 2 demonstrates, it is possible to find circumstances under which the model community is destabilised by the inclusion of dispersal. Figure 2a shows the eigenvalue spectrum for the model without spatial effects ($q = 0$, or equivalently $D_u = D_v = 0$, see Methods). All eigenvalues in Fig. 2a have negative real parts so we conclude that the equilibrium is stable for the model ecosystem without dispersal. In Fig. 2b we take into account dispersal and show the eigenvalue spectrum for a non-zero wavenumber. All other parameters are the same as in Fig. 2a. Now, an outlier eigenvalue strays over the imaginary axis, demonstrating that the equilibrium is unstable. For perturbations with higher wavenumbers, the outlier returns to the negative half-plane (Fig. 2c)—the equilibrium is stable with respect to perturbations of higher wavenumber. Figure 2d shows the real part of the rightmost eigenvalue ($\mathrm{Re}\,[\omega_{max}]$) as a function of the wavenumber $q$, highlighting the set of wavenumbers against which the equilibrium is unstable.

To understand the role of trophic structure in dispersal-induced instability, let us consider momentarily a model without statistical distinction between predator and prey species (like

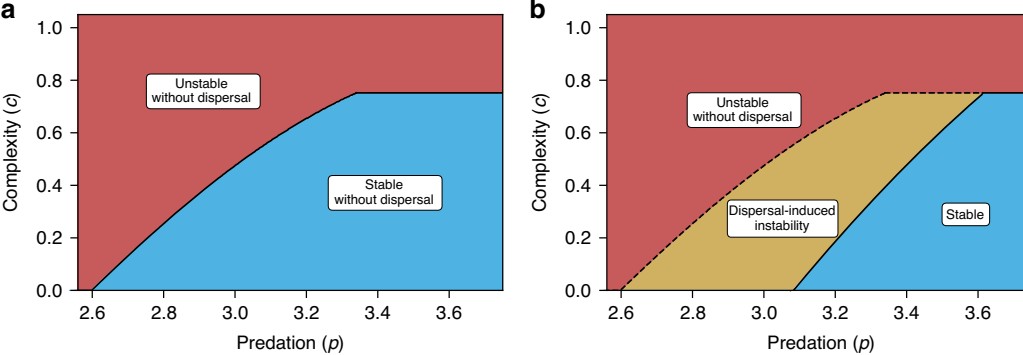

**Fig. 3 Dispersal as a destabilising influence.** Panel **a** shows the stability diagram for the non-spatial ecosystem model with trophic structure. The solid line is the upper bound on complexity $c$ for the equilibrium to be stable. Increasing complexity leads to instability. There is also a lower bound on the amount of predation $p$ required for the community to be stable (if $p$ is too small, then the equilibrium is unstable even for small values of the complexity). Stability for the spatial ecosystem with dispersal is shown in panel (**b**). In the blue region, the equilibrium is stable. In the yellow region, the equilibrium would be stable in a non-spatial model but dispersal in space induces instability. Crossing from the blue into the yellow region in panel (**b**) the ecosystem undergoes a Turing instability. In the red region in (**b**) the equilibrium is unstable both in the non-spatial and in the spatial ecosystem.

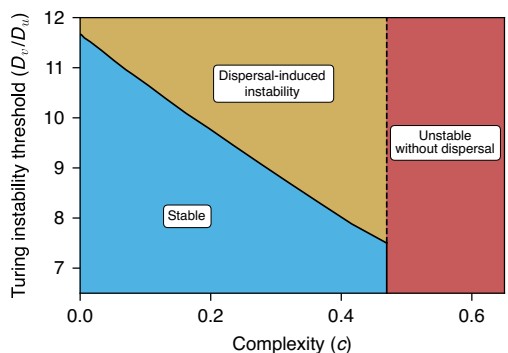

**Fig. 4 Complexity reduces the ratio of diffusion coefficients required for Turing instability.** In the yellow region, the community is unstable due to Turing instability. Conversely, a high value for the ratio of diffusion coefficients $D_v/D_u$ reduces the complexity that can be sustained in stable equilibrium. For sufficiently high complexity the non-spatial community becomes unstable (red area to the right of the vertical line). The equilibrium of the spatial ecosystem is then also unstable, irrespective of the diffusion coefficients.

May's original model[7]), and with a common diffusion coefficient $D$ for all species. The rightmost eigenvalue would have the following dependence on wavenumber: $\mathrm{Re}\,[\omega_{\max}] = \omega_0 - Dq^2$, where $\omega_0$ is a rightmost eigenvalue of the community stability matrix without dispersal ($q = 0$). In this simple case, $\mathrm{Re}\,[\omega_{\max}]$ is a purely decreasing function of $q$. If $\omega_0 < 0$ then $\mathrm{Re}\,[\omega_{\max}] < 0$ for all values of $q$. There can be no dispersal-induced instability here. The inclusion of trophic structure in our model leads to a maximum in $\mathrm{Re}\,[\omega_{\max}]$ at a non-zero value of $q$ (Fig. 2d) and, consequently, a finite band of non-zero wavenumbers against which the equilibrium is unstable. This is the essence of why dispersal in combination with tropic structure can promote instability.

In Turing's original work on chemical reaction systems[26], instability with respect to perturbations of a finite band of wavenumbers [as in Fig. 2d] signalled the formation of stable periodic patterns. The exact shape of these patterns is usually determined by non-linearities in the differential equations describing the reactions. Our model is valid only in the vicinity of the supposed equilibrium and, similar to May[7], we have not specified the nature of any non-linearities. We, therefore, do not

speculate for now about what might happen after the system has departed from the fixed point. We merely point out here the dispersal-induced instability of the equilibrium about which we have linearised.

**Modifying May's bound: stability with and without dispersal.** In order to further appreciate the effect that the inclusion of dispersal has on stability, we first consider the conditions under which the non-spatial ecosystem becomes unstable (see Methods). This enables us to study how May's bound on the complexity $c$ changes for our model, which has distinct predator and prey species. A stability plot is shown in Fig. 3a. The horizontal axis shows the average degree of predation $p = CN\mu_{vu}$ (see Methods). The vertical axis is the complexity parameter $c$. The solid line indicates the upper bound on the complexity: below the line the equilibrium is stable, above this line it is unstable.

We see from Fig. 3 that greater predation $p$ increases the amount of complexity $c$ that can be sustained in stable equilibrium by the ecosystem. Notably, in order to have stability at all in the model with trophic structure, there is a lower bound on the predation parameter $p$.

If we now include dispersal, the stability diagram changes [Fig. 3b]. In particular, the upper bound on the complexity can become lower than in the non-spatial system. This is because a new type of instability is now possible—the Turing instability. Thus, there are instances in which the model is stable without dispersal but unstable when dispersal is introduced (yellow area in Fig. 3b labelled 'dispersal-induced instability').

There are no situations in which an unstable equilibrium is stabilised by the combination of dispersal and trophic structure alone. However, previous work[25] has shown how the combination of spatial heterogeneity (in inter-species interactions) with dispersal can be stabilising. We comment on the effect of including spatial heterogeneity in our model in the Discussion and in the Supplementary Information (Section S14).

**How does complexity affect the Turing instability?** So far we have concerned ourselves with the effects of dispersal on complex ecosystems. We now ask the reverse question: spatial instability and pattern formation have been found in 'simple' models of ecosystems with a small number of species[36–39]. What are the effects of complexity on this Turing instability?

In general, Turing instabilities in simple systems typically occur when the diffusion coefficients of the 'activator' and 'inhibitor'

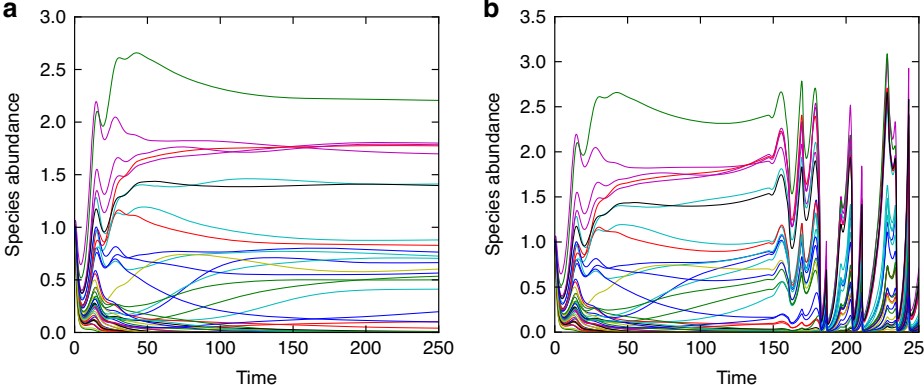

**Fig. 5 Dispersal can induce instability of feasible equilibria.** The prey species abundances in the non-spatial Levin-Segel ecosystem in panel (**a**) converge to a stable feasible equilibrium. In panel **b**, we simulate the ecosystem with the same parameters as in (**a**) but allowing for dispersal in space. Volatile behaviour is found in the spatial ecosystem, even though the non-spatial system would approach a stable equilibrium.

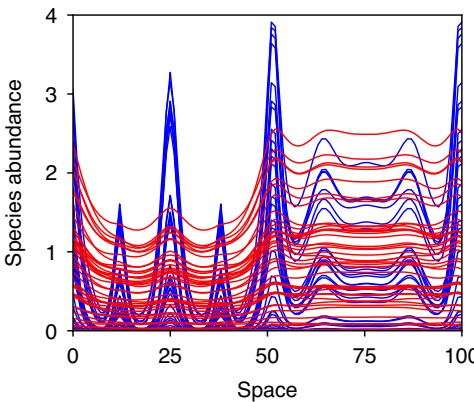

**Fig. 6 Distribution of species in space as a result of dispersal-induced instability.** The figure shows a snapshot of all $N = 100$ species abundances in a complex Levin-Segel ecosystem. The model parameters are the same as in Fig. 5b, which are given in full in the Supplementary Information. Predator species abundances are shown as red lines, prey species as blue lines. The most abundant species varies from place to place. This structure is not a static pattern as in conventional reaction-diffusion systems with Turing instability. Instead, the abundances change with time.

components are quite disparate—this is known as the 'fine-tuning issue' with the Turing mechanism[40,41]. The activating components in our system are the prey species, while predators play the role of the inhibitors. We would like to know what the effects of complexity are on the threshold ratio $D_v/D_u$ at which the Turing instability occurs. To answer this question, we compute this threshold for different values of the complexity parameter $c$.

The case $c = 0$ can be achieved by setting the width $\sigma$ of the distribution for the elements of the matrix **A** to zero. The matrix elements within a block are then identical to each other. The model thus reduces to a simple two-species predator–prey system. Increasing $c$ from zero introduces heterogeneity between the species within the blocks.

We find that complexity can lower the ratio of diffusion coefficients required for Turing instability (Fig. 4). For more complex ecosystems the Turing instability, therefore, sets in more easily. So, not only can complexity decrease the stability of non-spatial model ecosystems, but it can also reduce stability in spatial models. Conversely, increasing the ratio of diffusion coefficients $D_v/D_u$ reduces the complexity that can be sustained in stable equilibrium. As can be seen in Fig. 4, the upper bound on $c$ is lower when the disparity of dispersal rates is large.

**Spatial instability in a non-linear model with complexity.** The linear analysis we have focused on so far, although informative, has drawbacks. It only deals with the dynamics in the vicinity of a homogeneous equilibrium and it tells us nothing about how the ecosystem behaves in the long-run if the equilibrium is unstable. Further, one could object that the linear model is somewhat contrived and that it does not capture how a 'real' ecosystem constructs itself in equilibrium.

We now present simulation data of a complex ecosystem obeying non-linear Levin–Segel-type dynamics[31] (see Methods). In Fig. 5, we demonstrate that a dispersal-induced instability can occur in this model as well. Figure 5a shows a realisation of the dynamics without dispersal. The model ecosystem converges to an equilibrium composed only of surviving species. By definition, this is a feasible equilibrium[34].

Figure 5b shows the same model (with the same interaction matrix **A**) but now with dispersal. The abundances do not settle in the long run. Instead, they display quite erratic behaviour. We stress that this is different from the typical behaviour seen in two-species systems with a Turing instability. In such simple systems, the abundance of each species converges to a constant value eventually. The fixed-point value varies across space, generating a periodic pattern. The complex nature of the interactions in our model leads to the more complicated dynamics in Fig. 5b known as diffusion-induced chaos[42,43].

With that being said, at any point in time, one can take a snapshot of the spatial profile of the species abundances. An example is shown in Fig. 6. One finds some spatial structure for the abundances of prey species (blue lines) but this differs from classic Turing patterns, which are typically more periodic and regular. We note that the quickly diffusing predator species (red lines) have more smoothly undulating spatial profiles than the slowly diffusing prey species in this example.

## Discussion

The random-matrix approach to modelling ecosystems has developed substantially since May's original work. We contribute to this development with a model of a complex ecosystem which includes trophic structure and dispersal in space. These additional features change May's bound on complexity. How the bound changes tells us about the influence of the new model components on stability. For example, we find that predation acts as a stabilising influence (Fig. 3, and Sections S5 and S14 of the Supplementary Information).

Inspired by Turing's mechanism for pattern formation[26], we show how dispersal can be a destabilising factor in a complex

ecosystem. An equilibrium which would be stable in the non-spatial model can become unstable through dispersal. Such instability has been suggested as a mechanism for the observed patterns in natural ecosystems with small numbers of species involved[44–49]. Our work extends the consideration to a larger number of species and suggests the possibility of dispersal-induced instability in more complex communities.

Conversely, we observe that increased complexity can lower the threshold for the diffusion coefficients required for a Turing instability. So, not only does complexity reduce stability in a non-spatial model as was May's conclusion[7], but it also destabilises spatial models with dispersal. This is interesting especially in light of the so-called 'fine-tuning' issue with the Turing mechanism. The ratio of diffusion coefficients required for a Turing instability is usually large, making it hard to find experimental examples of Turing pattern formation[33,41,50,51]. Based on our findings, we speculate that dispersal-induced instability may be easier to observe in complex dynamical systems.

To demonstrate that dispersal-induced instability can also be seen in model ecosystems where the equilibrium is arrived at more naturally, we performed simulations of a complex ecosystem with Levin-Segel dynamics (Fig. 5). We found that this model also exhibits a dispersal-induced instability. Because of the complex nature of the interactions, the system does not reach a stable patterned state, which is normally characteristic of Turing instabilities in models with fewer species. Instead, one sees persistently volatile dynamics known as diffusion-induced chaos[42,43].

So as to highlight the destabilising effect of the combination of dispersal and trophic structure as clearly as possible, we were parsimonious with the inclusion of additional detail in the model. It is possible to relax some of the restrictions that we imposed without sacrificing the ability to perform the mathematical analysis. For example, variation in the autoregulation coefficients $d_\alpha$ and the dispersal coefficients $D_\alpha$ between species can be taken into account (in a similar way to ref. [52]). In Section S9 of the Supplementary Information, we show that the possibility of dispersal-induced instability remains despite this additional complication.

Provided that relatively mild conditions are met, we also demonstrate that our results still apply when the random matrix elements are drawn from a non-Gaussian distribution (see Section S7 in the Supplementary Information). More precisely, we show that the eigenvalue support only depends on the first and second moments of the distribution of the random matrix elements $A_{ij}^{\alpha\beta}$. This feature of the random matrix model is known as universality[53,54]. This is interesting from a theoretical point of view and also allows one to constrain the signs of the interaction coefficients within the different blocks in the interaction matrix (see Section S8 in the Supplementary Information). This means that it is possible to enforce strict loss-gain interactions between prey and predator species and to eliminate effects such as intra-guild predation. Again, the possibility of dispersal-induced instability remains.

In spatial models of ecosystems, dispersal is often implemented as migration between discrete patches rather than by diffusion in a continuous domain[27,28]. The set of interspecies interactions in the different patches can then be conceptualised as the edges of a multi-layer network[14,15]. Our results, which were derived using diffusive dispersal in continuous space, also continue to hold in such meta-ecosystems (see Sections S1C and S13 of the Supplementary Information).

A patched landscape is also a particularly convenient way to include spatial heterogeneity in the model. A recent study by Gravel et al.[25] demonstrated that dispersal can stabilise equilibria in complex ecosystems with spatial heterogeneity (we replicate

these findings with our method in Section S11 of the Supplementary Information). This study did not include trophic structure, which is a key aspect of our model. If spatial heterogeneity is combined with trophic structure, both the stabilising mechanism reported in[25] and the dispersal-induced instability at the focus of our work can be seen in the same model (see Section and S14 in the Supplementary Information and in particular Fig. S13).

The stabilising effect in ref. [25] and the destabilising mechanism we present can be viewed as somewhat separate. Dispersal-induced instability is associated with the outliers in the eigenvalue spectrum. The basis for the stabilising mechanism in ref. [25] is a reduction in the bulk eigenvalue spectrum (see Supplementary Information Sections S11 and S12). Which one of the effects takes precedence depends on the circumstances. A community with trophic structure and a significant predator-to-prey ratio of dispersal rates will be more likely to exhibit dispersal-induced instability. A community with significant spatial variation in interactions and consistently high dispersal rates across all species will be more likely to be subject to the stabilising effects of dispersal (for further discussion see Section S14 of the Supplement and Fig. S14 in particular).

One criticism levelled at May's model is that it is too simple and that perhaps through the inclusion of further aspects of natural ecosystems, the upper bound on the complexity could be eliminated. Our results do not support this hypothesis. Like May, we also find that there is always an upper limit on the complexity $c = \sigma^2 NC$ that an ecosystem can stably sustain, even when stabilising factors such as predation and spatial heterogeneity are taken into account. Other recent studies using random-matrix approaches arrive at similar conclusions. For example, Allesina and Tang[12] take into account more realistic food-web structures and still find upper limits on the number of interconnected species[13]. May's result, therefore, generalises to models capturing more aspects of 'real' communities in ecology. This prediction is supported by the observation of 'diversity regulation' in some ecosystems[55–59].

A final observation that we wish to convey is that making models for complex ecosystems more detailed introduces the opportunity for new types of instability. In May's original model, for example, the mean of the community matrix elements was zero. As a consequence, any one species is equally likely to benefit or suffer from the presence of another species. Mathematically, all eigenvalues then reside within one bulk region, and it is this bulk region that determines stability. Mutualism can be introduced through interaction coefficients which are positive on average, and competition through a negative average interaction[12]. This leads to additional outlier eigenvalues, which can make an equilibrium unstable even though it would otherwise be stable. Introducing the trophic structure can generate complex-conjugate pairs of outliers (Fig. 2a), allowing further opportunity for instability (Section S5 of the Supplementary Information). Dispersal, finally, leads to the possibility of a Turing-type instability. Overall, adding more details to the model of May tends to give rise to more ways in which equilibrium can become unstable.

## Methods

**Linear model.** We imagine that we find the ecosystem at a homogeneous equilibrium. Our model is concerned with the dynamics of small perturbations of the species abundances about this fixed point. The stability of the homogeneous fixed point is determined by whether or not these perturbations decay or increase with time. We write $u_i(x, t)$ and $v_j(x, t)$ for the perturbations of the prey and predator species abundances respectively at position $x$ and time $t$. These are the deviations away from the fixed point. There are $N_u$ prey species and $N_v$ predator species with $N = N_u + N_v$ species in total. We define the constants $\gamma_u = N_u/N$ and $\gamma_v = N_v/N$.

We assume that prey species diffuse at rate $D_u$ and predator species at rate $D_v$ in a spatially homogeneous environment. Similar to May[7], we also imagine that all

species in each group have equal and negative self-interaction. However, it is possible to relax the simplifying constraints of uniform self-interaction and diffusion coefficients (see Section 9 in the Supplementary Information and ref. [52]). The probability that a particular pair of species interact with one another is $C$. This parameter is known as the 'connectance'[7]. The effect of a change in the abundance of species $j$ on species $i$, where $j$ belongs to trophic block $\beta$ and $i$ belongs to $\alpha$, is $A_{ij}^{\alpha\beta}$ [$\alpha, \beta \in \{u, v\}$]. The linearised reaction-diffusion equations thus take the following form

$$\frac{\partial u_i}{\partial t} = D_u \frac{\partial^2 u_i}{\partial x^2} - d_u u_i + \sum_{k=1}^{N_u} A_{ik}^{uu} u_k + \sum_{j=1}^{N_v} A_{ij}^{uv} v_j,$$
$$\frac{\partial v_j}{\partial t} = D_v \frac{\partial^2 v_j}{\partial x^2} - d_v v_j + \sum_{i=1}^{N_u} A_{ji}^{vu} u_i + \sum_{k=1}^{N_v} A_{jk}^{vv} v_k.$$
(3)

If species $i$ and $j$ are non-interacting, $A_{ij}^{\alpha\beta} = A_{ji}^{\beta\alpha} = 0$. If the two species interact, then $A_{ij}^{\alpha\beta}$ and $A_{ji}^{\beta\alpha}$ are drawn from a joint Gaussian distribution with means $\overline{A_{ij}^{\alpha\beta}} = \mu_{\alpha\beta}$ and $\overline{A_{ji}^{\beta\alpha}} = \mu_{\beta\alpha}$. The elements in the random matrix have variance $\sigma^2$

$$\overline{(A_{ij}^{\alpha\beta} - \mu_{\alpha\beta})^2} = \sigma^2,$$
(4)

and they are correlated according to

$$\overline{(A_{ij}^{\alpha\beta} - \mu_{\alpha\beta})(A_{ji}^{\alpha'\beta'} - \mu_{\alpha'\beta'})} = \Gamma_{\alpha\beta\alpha'\beta'}\sigma^2.$$
(5)

All other correlations are set to zero. Each of the model parameters can be interpreted ecologically. We define and interpret the complexity $c = CN\sigma^2$ in the main text. The interaction means $\mu_{uu}$ indicates the degree to which different prey species cooperate (if $\mu_{uu} > 0$) or complete (if $\mu_{uu} < 0$). The coefficient $\mu_{vv}$ has a similar interpretation for predators. The means of the off-diagonal blocks $\mu_{uv} < 0$ and $\mu_{vu} > 0$ indicate the degree to which (on average) prey species suffer and predator species gain from predator–prey interactions. The parameters $\Gamma_{\alpha\beta\alpha'\beta'}$ describe the correlations between interaction coefficients. The only non-zero entries are taken to be $\Gamma_u \equiv \Gamma_{uuuu}$, $\Gamma_v \equiv \Gamma_{vvvv}$ and $\Gamma_{uv} \equiv \Gamma_{uvvu}$. That is, only elements which are diagonally opposite one another in $\mathbf{A}$ are correlated. A positive value of $\Gamma_{\alpha\beta\alpha'\beta'}$ indicates that if one species benefits more than average from interaction, the other species involved does so as well. The opposite is true if $\Gamma_{\alpha\beta\alpha'\beta'}$ is negative.

Taking the Fourier to transform with respect to the spatial coordinate $x$ of Eq. (3), we arrive at dynamical equations (see Supplementary Information Section S1) for disturbances of wavenumber $q$ in the abundances of the various species (the wavenumber is related by $q = 2\pi/\lambda$ to the wavelength $\lambda$). We denote the combined vector of the Fourier transforms of species abundances by $\tilde{\mathbf{X}}_q = (\cdots, \tilde{u}_i(q,t), \cdots, \tilde{v}_j(q,t), \cdots)$, and arrive at the more compact matrix equation

$$\dot{\tilde{\mathbf{X}}}_q = \mathbf{M}_q \tilde{\mathbf{X}}_q.$$
(6)

The matrix entry $\left(\mathbf{M}_q\right)_{ij}^{\alpha\beta}$ tells us what the effect of a disturbance of wavenumber $q$ in species $j$ (belonging to trophic block $\beta$) is on species $i$ (belonging to trophic block $\alpha$).

The vector $\mathbf{X}_q$ has dimension $N = N_u + N_v$ and is arranged such that the first $N_u$ elements are the Fourier-transformed abundances $\tilde{u}_i(q, t)$ and the last $N_v$ elements are the $\tilde{v}_j(q, t)$. The matrix $\mathbf{M}_q$ is depicted in Fig. 1. It is divided into blocks due to the trophic structure of the community. Its three contributions are a diagonal diffusion matrix, a diagonal self-interaction matrix and a random interaction matrix, whose variance and correlations are given in Eqs. (4) and (5). The indices $\alpha$ and $\beta$ correspond to the different blocks and $i$ and $j$ correspond to the position within the block. If the matrix $\mathbf{M}_q$ has eigenvalues with positive real parts for any value of $q \geq 0$, the disturbances $u_i$ and $v_i$ will grow with time, indicating an unstable equilibrium.

By setting $q = 0$ in Eq. (6), one recovers Eq. (3) with the diffusion term removed. Focusing on $q = 0$ in our model (or equivalently setting $D_u = D_v = 0$) thus allows one to study the stability of a non-spatial model ecosystem with trophic structure. Further details can be found in Sections S1 and S5 in the Supplementary Information.

We note that a similar form for the matrix $\mathbf{M}_q$ can be found for a model of a meta-ecosystem with dispersal between discrete patches (similar to ref. [27,28]) instead of in continuous space (see Sections S1 C and S13 of the Supplementary Information).

The values of the model parameters used in the figures are given in full in Section S6 of the Supplementary Information.

**Calculation of the boundary surrounding the bulk of the eigenvalues.** The vast majority of the eigenvalues of the random matrices $\mathbf{M}_q$ reside within one or two 'bulk' regions of the complex plane. To determine stability we need to know if there are bulk eigenvalues with positive real parts. Identifying the boundaries of the regions containing the eigenvalues is sufficient for this purpose. Our calculation uses methods originally developed in statistical physics and follows lines similar to

those of refs. [60,61]. This approach converts the problem into the evaluation of a 'potential' related to the eigenvalue density. This potential, in turn, can be expressed as a high-dimensional integral, which is carried out using the saddle-point method. A brief summary of the context of this approach in the wider literature is given Section S2 A of the Supplementary Information. Full details of the calculation are given in Sections S2 B and S2 C.

We write $\omega = \omega_x + i\omega_y$ for the eigenvalues of $\mathbf{M}_q$. The general expression for the boundary surrounding the bulk of the eigenvalues ($\omega_y$ as a function of $\omega_x$) is given by the simultaneous solution of the following equations (see Supplementary Information Eq. (S81))

$$\sum_\alpha \gamma_\alpha |\chi_\alpha|^2 - \frac{1}{c} = 0,$$
$$-(\omega_x + i\omega_y + d_\alpha + q^2 D_\alpha)\chi_\alpha + c \sum_\beta \Gamma_{\alpha\beta}\gamma_\beta \chi_\alpha \chi_\beta + 1 = 0.$$
(7)

We note the free index $\alpha$ in the second of these equations ($\alpha \in \{u, v\}$). This is therefore a system of three coupled equations. One first eliminates the auxiliary variables $\chi_\alpha$, and then expresses $\omega_y$ in terms of $\omega_x$. This results in the red curves in Fig. 2.

The solution simplifies in several special cases, which we exploit to provide explicit stability criteria analogous to May's bound (Section S5 of the Supplementary Information).

**Calculation of the outlier eigenvalues.** In addition to the bulk eigenvalues, the stability matrix can have isolated outlier eigenvalues. If any of these outliers have a positive real part, the equilibrium is unstable. Their position in the complex plane is calculated along the lines of ref. [62]. Details can be found in the Supplementary Information Section S3. The outlier eigenvalues are given by the complex values $\omega$ satisfying the following equation (see Supplementary Information Eq. (S82))

$$\left[\gamma_u NC\mu_{uu} - \frac{1}{\chi_u(\omega)}\right]\left[\gamma_v NC\mu_{vv} - \frac{1}{\chi_v(\omega)}\right] - (NC)^2 \gamma_u \gamma_v \mu_{uv}\mu_{vu} = 0,$$
(8)

The auxiliary quantities $\chi_\alpha(\omega)$ in this relation satisfy

$$-1 = -(\omega + d_u + q^2 D_u)\chi_u + c\Gamma_u \gamma_u \chi_u^2 + c\Gamma_{uv}\gamma_v \chi_v \chi_u,$$
$$-1 = -(\omega + d_v + q^2 D_v)\chi_v + c\Gamma_{uv}\gamma_u \chi_u \chi_v + c\Gamma_v \gamma_v \chi_v^2,$$
(9)

subject to the condition

$$\sum_\alpha \gamma_\alpha |\chi_\alpha(\omega)|^2 < \frac{1}{c}.$$
(10)

Eqs. (8) and (9) need to be solved simultaneously, subject to Eq. (10). If there are no solutions then there are no outliers. In special cases, the above expressions can be simplified, and explicit stability criteria can be found (Section S5 in the Supplementary Information).

**Finding the threshold for instability.** Instability can occur in one of several different ways: (1) The bulk region of eigenvalues for $q = 0$ can cross into the positive half-plane; (2) one of the outlier eigenvalues for $q = 0$ can cross the imaginary axis; (3) an outlier eigenvalue for $q \neq 0$ can stray into the positive half-plane. We have not observed any circumstances under which the bulk crosses the imaginary axis for non-zero $q$ where it does not for $q = 0$. The eigenvalues must have negative real parts for all $q$ in order for the equilibrium to be stable in the spatial system. This includes $q = 0$. Cases (1) and (2) therefore indicate instabilities occurring both in the non-spatial and the spatial ecosystem. In case (3) the spatial system is unstable, but the non-spatial system remains stable. In each of these cases, the threshold for instability is found by identifying sets of parameters for which either the boundary for the bulk eigenvalues touches the imaginary axis (case 1) or where the outlier eigenvalues touch the imaginary axis (cases 2 and 3).

This can be done using the analytical results for the spectrum of eigenvalues (Section S5 of the Supplementary Information), leading to the results shown in Figs. 3 and 4.

**Simulating the non-linear model.** Results in Figs. 5 and 6 are from numerical integration of the Levin–Segel-type model[31],

$$\frac{\partial u_i}{\partial t} = D_u \frac{\partial^2 u_i}{\partial x^2} + u_i \left[a - u_i + \sum_{k \in u} A_{ik}^{uu} u_k + \sum_{j \in v} A_{ij}^{uv} v_j\right],$$
$$\frac{\partial v_j}{\partial t} = D_v \frac{\partial^2 v_j}{\partial x^2} + v_j \left[-v_j + \sum_{i \in x} A_{ji}^{vu} u_i + \sum_{k \in v} A_{jk}^{vv} v_k\right].$$
(11)

where $a > 0$ is a constant. To integrate these equations numerically, the diffusion terms are discretised. The integration is then carried out using the Runge–Kutta (RK4) method[63].

**Further variations on the model.** The flexibility of our analytical approach allows us to include additional features to our fairly austere model. For example, in Sections S7 and S8 of the Supplementary Information, we demonstrate the universality of our theoretical results. That is, we show that the matrix elements need not be drawn from

a Gaussian distribution for our results to apply. Similar to ref. [52], variation in the self-interaction and diffusions coefficients ($d_\alpha$ and $D_\alpha$ respectively) can also be taken into account (Section S9 in the Supplementary Information). In Section S13 of the Supplementary Information, we show that the dispersal-induced instability persists on a landscape of discrete patches (as opposed to diffusion in a continuous space) and when dispersal is non-local. These extensions highlight the robustness of the dispersal-induced instability in complex ecosystems and the versatility of the analytical formalism.

**Inclusion of spatial heterogeneity in the interaction coefficients**. In analysing Eqs. (3) and the stability matrix in Eq. (2) we have assumed that the interaction coefficients $A_{ij}^{\alpha\beta}$ are the same at every point in space. In order to model spatial heterogeneity, we extend the set up along the lines of ref. [25] and imagine that dispersal takes place on a set of discrete patches indexed by their position $x$, similar to meta-population models[28] (see Section S10 in the Supplementary Information). The interaction matrix $A_{ij;xx'}^{\alpha\beta}$ then has two layers of block structure: one indicating the trophic structure as before and the second representing a location in space. Multilayer networks have previously been used to encapsulate similar structures in ecological communities[14,15].

Adapting the prior calculation, the regions in the complex plane containing the bulk and outlier eigenvalues can also be computed for the model with spatial heterogeneity (Section S11 in the Supplementary Information). The model in the main text and that of ref. [25] are special cases of this setup. In particular, we can also recover the eigenvalue support and stability criteria of ref. [25]. In Section 14 of the Supplementary Information, we show that the mechanism leading to dispersal-induced instability and the stabilising mechanism of ref. [25] can coexist in the same model. We also discuss the factors that determine whether dispersal acts to stabilise or destabilise equilibria of complex ecosystems.

**Reporting summary**. Further information on research design is available in the Nature Research Reporting Summary linked to this article.

## Data availability

The data in Figs. 2–6 is generated using the codes in the code availability statement. The data are also available upon reasonable request to the corresponding author.

## Code availability

Codes for Figs. 2–6[64] are available from the following link https://doi.org/10.5281/zenodo.4068257. They are written using Mathematica 12 and Python v3.8. Python packages matplotlib, numpy and scipy were used. The codes for producing the figures in Supplementary Information are available upon reasonable request to the corresponding author.

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

## Acknowledgements

J.W.B. thanks the Engineering and Physical Sciences Research Council (EPSRC) for funding (PhD studentship, EP/N509565/1). J.W.B. and T.G. are grateful for funding from the Spanish Ministry of Science, Innovation and Universities, the Agency AEI and FEDER (EU) under grant PACSS (RTI2018-093732-B-C22). T.G. acknowledges partial financial support from the Maria de Maeztu Programme for Units of Excellence in R&D (MDM-2017-0711).

## Author contributions

J.W.B. designed the study, contributed to discussions guiding the work, carried out mathematical calculations, performed simulations and wrote the paper. T.G. designed the study, contributed to discussions guiding the work and wrote the paper. Both authors approved the final paper for submission.

## Competing interests

The authors declare no competing interests.
