## [Peer Review File · Nature Communications]

Reviewers' Comments:

Reviewer #1:

Remarks to the Author:

The authors present a manuscript about instability in complex ecosystems. Although it is not a novel problem, the authors present a very thorough approach including dispersal in the species. Starting from the original proposals by Robert May they include diffusion as a mobility that can even be eventually generalized to a discrete patch of sites.

The paper is very well written and presented.

The methodology is sound, based on advanced techniques of statistical physics on random matrices. And the effect of mobility is shown from a theoretical and from a computational point of view. Simulations agree very well with the predictions.

The work could be eventually extended in some directions, but the authors plan is to focus on the simple effect of mobility showing that, indeed, it enlarges the instability region.

They have not considered the effect of dispersion in the diffusion or in the selfinteraction term.

Although it represents a strong constraint, if the focus is on the dispersion the goal is maintained. In any case, some comment on this and perhaps some intuitive trend could be useful.

Also, the presence of statistical correlations in the interaction matrix could be eventually considered and at least exposed.

One of the things I miss in the manuscript is some references to works about where the diffusion takes place in a discrete patch landscape, such that diffusion is considered to occur on a complex multiplex network. In this case multiplex is understood as a multilayer network with a different connectivity pattern in each layer for predators and preys.

As stated before I consider the paper to represent a consistent approach to the stability conditions and only a few additions should be necessary, considerations about other directions of diversity and references on multiplex approaches.

Reviewer #2:

Remarks to the Author:

The manuscript of Baron and Galla investigates the local asymptotic stability of random community matrices with two distinct trophic levels and a linear continuum of potential habitats. In Fourier space, the differential diffusion terms transform into matrices acting on spatial Fourier components, while leaving self-regulation and species interaction terms untouched. The Authors find that the added complexity to May's original random matrix, instead of stabilizing them, introduces extra opportunities for destabilization. In particular, there is a range of spatial wave numbers for which the leading eigenvalue of the corresponding community matrix has positive real part.

The manuscript is clear and generally well-written. It has both a biological and a methodological aspect, and I will comment on both below.

Biologically, the message is that the inclusion of space and distinct trophic levels can destabilize communities. It might be worth decoupling the two, for the sake of illustration. Do the same results hold with the same spatial model but with a single, simple elliptic interaction matrix? And vice versa?

The formalism described in the Supplement ought to apply to these cases as well, and even in a simpler form, so hopefully any argument can be supported analytically.

Disentangling the effects of space and trophic structure would also be important because, as the Authors point out, the study of Gravel et al. (2016 Nat Commun) arrives at the opposite conclusion with a very similar model that does not have the two distinct trophic levels. My feeling is that the two models ought to give the same result if, instead of connecting every patch to every other one, we set up a linear chain of discrete patches with migration only between every adjacent ones. While my purpose is not to overwhelm the Authors with extra requests for analyses, this should be easy to check numerically.

If the predictions do not match up after all, that might be because in the discrete case, the number and density of patches must be finite, restricting the possible wave numbers. This brings me to another point: it is not obvious from Figure 2d whether the wave numbers corresponding to instability are "high" or "low", compared to some natural scale of the system - this would be good to clarify somewhere. (If the given q values are in practice pathologically low, then the problem of destabilization is largely academic.)

But if the predictions of the two models are in fact compatible (which, again, I would expect to be the case), then I would also rephrase the Discussion. I would not say that results are sensitive to the way dispersal is modeled, or that the Authors' way of modeling space is more realistic. (In any case, it is debatable whether a continuum of fully intact communities along a line extending from minus to plus infinity is more realistic than a discrete number of equally connected ones.) Instead, I would say that the tension between the two results is superficial, and hinges on the geometry of the landscape. If every patch is reasonably accessible from every other patch, then space may stabilize. Whereas if space consists of a linear chain of many patches, there is a destabilizing effect.

Lastly, the way the trophic levels are implemented is somewhat suspect. The problem is not with having two strict levels. Rather, it seems that feeding interactions are present even within a single level, since the interaction matrices within a level are generated as random matrices drawn from Gaussian ensembles. In principle therefore, nothing prevents two prey species having a (+,-) relationship with one another (intraguild predation). Is this indeed the case? If so then I would either think more about the interpretation of the blocks (which are not necessarily trophic levels *sensu stricto*), or else restrict entries within levels to exclude trophic interactions.

About the methodology, I wanted to bring up two points. The first concerns universality. While the random parts of the matrices are always drawn from Gaussian distributions, it is implied at several places that the results are universal (i.e., under mild conditions, only the mean, variance, and correlation structure of the distribution matters, not its precise shape). Is that really so? Universality has been proven for circular and elliptic matrices, but these are difficult proofs and, to my knowledge, have not yet been generalized to more complicated random matrices. Is universality a theorem here, or more of a numerically supported and highly reasonable conjecture? I would state which it is either way. Also, if it is the latter, then I would do a few quick numerical studies, drawing matrix entries from two different distributions with the same means, variances, and covariances, to verify that they lead to the same asymptotic eigenvalue distributions.

The second point is about how the method developed by the Authors connects to existing ones. In my mind, the methodological contribution is as important (if not more so) than the biological one, since it enables one to determine the spectral density and spectral abscissa of a new class of matrices. But then, it would help to carefully point out how and where the method differs from, and fits in with,

seemingly very similar contributions. For instance, Allesina et al. (2015 Nat Commun) and Grilli et al. (2017 Nat Commun) discuss in depth how to evaluate the spectral density functions of matrices with block structure (corresponding to the two trophic levels here), and Barabás et al. (2017 Nature Ecol Evol) discuss the resolvent method and how to take unequal diagonal entries into account. While the Supplement is didactically very well written, it might be worth putting more emphasis on what the methodological contribution of this manuscript is, in relation to its predecessors. This would make it easier for people to know under what circumstances its results can and cannot be used, and whether multiple different approaches could be used to reach the same conclusions.

Additionally, a few minor points:

- l.21-23: I would try to avoid falling in the trap of giving a naïve account of the supposed naïveté of the cited works. Such overwhelming confidence about one's own intuition is definitely absent from e.g. MacArthur (1955 Ecology). I suggest starting the paper with "Providing a firm counterpoint to the view that greater ecosystem complexity promoted their stability [1-6], Robert May used ..." (or something to that effect).
- l.56-57: "variation in interaction strength" - that was surely one thing that May's original model did capture, or am I missing something?
- Fig. 2 caption: I would write "rightmost eigenvalue" instead of "most unstable eigenvalue".
- l.250: like -> likely
- l.272: This is not actually a necessary condition for stability. When the correlation between the (i,j)th and (j,i)th entries of the matrix is strongly negative, then it is possible for at least some diagonal entries to be positive and for the system to remain stable (see Barabás et al. 2017 Nature Ecol Evol for details).

Sincerely,
Gyuri Barabás

Manuscript ID NCOMMS-20-09236

Title of submission:

‘Dispersal-induced instability in complex ecosystems’
by Joseph W Baron and Tobias Galla

Reply to Referees and changes made to the manuscript

(Dated: September 2, 2020)

We thank both Reviewers for their insightful comments and for the time spent preparing their reports. Their suggestions were helpful to us in improving the manuscript and prompted us to think further about different aspects of our work. Prompted by the comments we have carried out further analytical calculations and numerical work. We believe these additional analyses make our characterisation of the effects of dispersal on the stability of complex ecosystems more complete and well-rounded.

Below we reply to the Reviewers’ comments point-by-point, highlighting the corresponding additions and changes we have made in the manuscript and supplement. We have numbered the different points made by each Referee for clarity and so as to be able to cross-reference between the reports.

The references to literature in this document are labelled with a prefix ‘R’ and are listed at the end of this response. For the sake of clarity, we sometimes highlight the label of the reference in the revised manuscript. Unless explicitly stated, all references to equations, figures and sections in the manuscript are with respect to the revised submission.

To assist the Referees we include a version of the manuscript in our submission in which the principal changes are highlighted.

I. REPLY TO REVIEWER 1

(1) The authors present a manuscript about instability in complex ecosystems. Although it is not a novel problem, the authors present a very thorough approach including dispersal in the species. Starting from the original proposals by Robert May they include diffusion as a mobility that can even be eventually generalized to a discrete patch of sites. The paper is very well written and presented. The methodology is sound, based on advanced techniques of statistical physics on random matrices. And the effect of mobility is shown from a theoretical and from a computational point of view. Simulations agree very well with the predictions.

Response:

We thank the Reviewer for their positive feedback. We do of course agree that investigating the factors that stabilise or destabilise complex ecosystems has a long tradition in theoretical ecology, sparked by the work by May in the 1970s. Surprisingly though, the effects of dispersal on random-matrix models has received relatively little attention so far. The work by Gravel et al [R1] (Ref. [25] in the revised manuscript) is one notable exception. We believe, our work is the first to systematically investigate when dispersal-induced instabilities might occur in complex ecological communities with trophic structure.

(2) *The work could be eventually extended in some directions, but the authors plan is to focus on the simple effect of mobility showing that, indeed, it enlarges the instability region. They have not considered the effect of dispersion in the diffusion or in the self-interaction term. Although it represents a strong constraint, if the focus is on the dispersion the goal is maintained. In any case, some comment on this and perhaps some intuitive trend could be useful.*

Response:

We thank the Referee for bringing this up. In the original submission we had indeed used uniform self-interaction coefficients for all predator species, and similarly the self-interaction for prey species was also uniform. These were denoted d_u for prey, and d_v for predator. Similarly, all species in each group had the same diffusion constant (labelled D_u and D_v respectively). We deliberately modified May’s original model sparingly, as our aim was to focus on the effects of diffusion in space combined with trophic structure. This minimal extension allowed us to highlight how dispersal affects stability without the potential obfuscation arising from other additions to the model.

We agree with the Referee, introducing dispersion (variation) of self-interaction and/or diffusion coefficients within each group is an ecologically worthwhile extension of the model. This is also interesting mathematically. We note that the comment relates to item (6) in Referee 2’s report.

In response to these suggestions we have extended the theoretical analysis to the case of varying diagonal elements. Our analytical formalism is sufficiently versatile to allow this, but requires a non-trivial modification. This involves averaging over the distribution of diagonal elements in a way similar to Ref. [R2], whose results we are able to recover. This analysis is presented in Sec. S9 of the revised Supplement (‘Variation in diagonal elements’). We find that the dispersal-induced instability remains possible in face of variation in the self-interaction or diffusion coefficients.

Changes made to the manuscript:

In addition to the new Section S9 in the Supplement, we comment on the possibility of variation in the diagonal matrix elements (along with the relaxation of other model assumptions) in the Discussion and Methods sections of the revised manuscript.

(3) *Also, the presence of statistical correlations in the interaction matrix could be eventually considered and at least exposed.*

Response:

The Referee is right in that we have not allowed for the most general case of correlations in the interaction matrix. Given that the interaction matrix in our model has block-structure there is, in principle, a large number of correlations one could consider. The explicit inclusion of such correlations in the model would lead to a larger set of model parameters. This, in turn, would make it difficult to analyse the model comprehensively and to pin-point the key effects of dispersal. This is why we opted to modify Robert May’s original model only in moderation.

We do stress however that our model allows for correlations between elements A_{ij}^{uu} and A_{ji}^{uu} and, separately, for A_{ij}^{vv} and A_{ji}^{vv} . These correlations are quantified by the coefficients Γ_u and Γ_v respectively. In our analytical calculations we also allow for correlations of A_{ij}^{uv} and A_{ji}^{vu} , which are

specified by Γ_{uv} . This is described in the text immediately after Eq. (S2) in the Supplementary Material. We agree though that this could have been made clearer in the main manuscript.

Changes made to the manuscript: In response to the Reviewer’s comment, we now highlight the correlations explicitly in the main text after Eq. (2).

(4) One of the things I miss in the manuscript is some references to works about where the diffusion takes place in a discrete patch landscape, such that diffusion is considered to occur on a complex multiplex network. In this case multiplex is understood as a multilayer network with a different connectivity pattern in each layer for predators and preys.

Response and changes made:

The Referee is right, there is a substantial body of work on models of ecosystems in which diffusion takes place between a discrete set of patches, notably Hanski’s work on meta-populations [R3, R4]. This work is now highlighted in the Introduction and Discussion sections. The model in the main text is first presented in continuous space but our analysis would not materially change if we had chosen a discrete-space setting (this is discussed in Sections S1 C and S13 of the revised Supplement). We now also highlight this more clearly in the ‘Linear model’ heading in Methods, where we have added a brief comparison to Hanski’s work.

There has also been more recent work conceptualising the space-varying interactions between different species in an ecosystem as edges of a multilayer network (Refs. [R5, R6], for example). We agree with the Referee that it would have been good to comment on this, along with suitable references. We do this in the Discussion of the revised manuscript. This is also related to Referee 2’s comment (3) below concerning the work by Gravel et al. [R1]. In response to this comment by Referee 2, we have carried out a more detailed analysis of dispersal-induced instability in a discrete-patch model with spatially heterogeneous interactions between species. This set-up can be thought of as a multilayer network of different interactions. This is now briefly discussed in ‘Inclusion of spatial heterogeneity in the interaction coefficients’ in Methods.

(5) As stated before I consider the paper to represent a consistent approach to the stability conditions and only a few additions should be necessary, considerations about other directions of diversity and references on multiplex approaches.

Response:

We thank the Reviewer again for their time, their helpful report and for their positive comments on our work. As can be seen from our responses to items (1)–(4) we agree with all points and we have made changes accordingly.

II. REPLY TO REVIEWER 2 [DR BARABÁS]

(1) The manuscript of Baron and Galla investigates the local asymptotic stability of random community matrices with two distinct trophic levels and a linear continuum of potential habitats. In Fourier space, the differential diffusion terms transform into matrices acting on spatial Fourier components, while leaving self-regulation and species interaction terms

untouched. The Authors find that the added complexity to May's original random matrix, instead of stabilizing them, introduces extra opportunities for destabilization. In particular, there is a range of spatial wave numbers for which the leading eigenvalue of the corresponding community matrix has positive real part. The manuscript is clear and generally well-written. It has both a biological and a methodological aspect, and I will comment on both below.

Response:

We thank Dr Barabás for his thoughtful and detailed report, and we are happy to see that his overall assessment of our work is positive. The comments Dr Barabás raised are all valid, and responding to them has helped us improve the quality of the manuscript. We reply to the queries in detail below.

(2) Biologically, the message is that the inclusion of space and distinct trophic levels can destabilize communities. It might be worth decoupling the two, for the sake of illustration. Do the same results hold with the same spatial model but with a single, simple elliptic interaction matrix? And vice versa? The formalism described in the Supplement ought to apply to these cases as well, and even in a simpler form, so hopefully any argument can be supported analytically.

Response and changes made in the manuscript:

We thank the Referee for prompting us to think in more detail about the different components of the model. The suggestion to disentangle the different components is indeed very useful, and Dr Barabás is right, the extended formalism does indeed cover models with trophic structure and no dispersal, and vice versa. Exploiting this to address the comment has helped us to develop a more rounded picture of the mechanism of dispersal-induced instability.

We can answer the first part of the Referee's question in the negative, dispersal-induced instability of the kind we discuss here is not found for simple elliptic random matrices. Intuition can be drawn from Turing's original work [R7]. Turing patterns arise in chemical systems due to a combination of short-range activation and long-range inhibition. If there is only one single type of 'chemical' then dispersal will act to homogenise, and a Turing instability cannot occur.

In the context of an ecosystem with no trophic structure this can also be seen as follows. In absence of dispersal, the spectrum of the interaction matrix is an ellipse in the complex plane with a possible outlier. If a uniform diffusion term is added to the model (with diffusion constant D), then for each wavenumber q the eigenvalues are shifted by an amount $-Dq^2$ along the real axis. Therefore, if all eigenvalues for $q = 0$ have negative real part, then the eigenvalues for $q \neq 0$ must also be in the negative half plane. Hence no dispersal-induced instability can occur for matrices with simple elliptic spectra. To see dispersal-induced instability, a non-monotonic dependence of the real part $\text{Re}[\omega_{\max}]$ of the leading eigenvalue on q is required (see Fig. 2d in the main text). This becomes possible through the introduction of slowly diffusing prey species and more quickly diffusing predator species. We have added a note to this effect in the revised Results Section 'Eigenvalue Spectra'.

The answer to the converse question is less clear-cut. The effects of introducing trophic structure in a non-spatial system depend on the detailed circumstances. To illustrate this with an example, we start from an interaction matrix with trophic structure, i.e. a matrix of the type shown in Fig.

1 of the main text (but with no dispersal). We then consider a corresponding ‘non-trophic’ random matrix, in which all interaction coefficients are drawn from the same distribution (no distinction between prey and predator blocks). We imagine that this distribution is such that it has mean, variance and correlations reflecting the overall statistics of the trophic matrix, aggregated over all four blocks. The resulting non-trophic matrix has an elliptic eigenvalue spectrum and at most one outlier.

We now focus on the specific example $\mu_{uu} = -\mu_{vv}$, $\mu_{uv} = -\mu_{vu}$, $\Gamma_{\alpha\beta} = 0$ for the system with trophic structure. For equal numbers of predator and prey species, the mean of all (off-diagonal) matrix elements is then $\bar{\mu} = 0$. This means that the elements of the non-trophic matrix have zero average, and therefore there are no outlier eigenvalues. The variance and diagonal elements can be chosen such that the non-trophic system is stable.

The expression for the outliers in the corresponding trophic case is given in Eq. (S93) of the revised Supplement. With the trophic structure, the system may or may not have outlier eigenvalues that cross the imaginary axis, depending on the coefficients $\mu_{\alpha\beta}$. More precisely, in Eq. (S93) there is a lower bound on the amount of predation ($p = NC\mu_{vu} = -NC\mu_{uv}$) required for stability. If p is below the threshold then the trophic system is unstable, if p is above the threshold it is stable. This also applies for more general parameters, as is illustrated in Fig. 3a of the main text. So, adding trophic structure to a random matrix with elliptic spectrum can destabilise a non-spatial model, but it does not have to.

To conclude our answer to item (2), we emphasise that our intention was to highlight that dispersal can have a destabilising effect on communities with trophic structure which otherwise are stable (part of the blue region in Fig. 3a becomes the unstable yellow region in Fig. 3b in the main text). In this context, we now emphasise more clearly the role played by trophic structure in introducing the lower bound on predation. This can be found under the heading ‘Modifying May’s bound’ in Results.

3 (a) Disentangling the effects of space and trophic structure would also be important because, as the Authors point out, the study of Gravel et al. (2016 Nat Commun) arrives at the opposite conclusion with a very similar model that does not have the two distinct trophic levels. My feeling is that the two models ought to give the same result if, instead of connecting every patch to every other one, we set up a linear chain of discrete patches with migration only between adjacent ones. While my purpose is not to overwhelm the Authors with extra requests for analyses, this should be easy to check numerically.

(b) If the predictions do not match up after all, that might be because in the discrete case, the number and density of patches must be finite, restricting the possible wave numbers. This brings me to another point: it is not obvious from Figure 2d whether the wave numbers corresponding to instability are "high" or "low", compared to some natural scale of the system - this would be good to clarify somewhere. (If the given q values are in practice pathologically low, then the problem of destabilization is largely academic.)

(c) But if the predictions of the two models are in fact compatible (which, again, I would expect to be the case), then I would also rephrase the Discussion. I would not say that results are sensitive to the way dispersal is modeled, or that the Authors’ way of modeling space is

more realistic. (In any case, it is debatable whether a continuum of fully intact communities along a line extending from minus to plus infinity is more realistic than a discrete number of equally connected ones.) Instead, I would say that the tension between the two results is superficial, and hinges on the geometry of the landscape. If every patch is reasonably accessible from every other patch, then space may stabilize. Whereas if space consists of a linear chain of many patches, there is a destabilizing effect.

Overall response to 3(a)–(c):

The Referee’s comments centre on the relation of our work to that by Gravel, Massoli and Leibold (GML) [R1]. We first respond to these items summarily, before we then address the specifics of each point in turn.

GML consider a model defined on discrete patches in which the interaction matrix can vary from patch to patch (spatial heterogeneity). They find that the combination of dispersal and spatial heterogeneity can act to stabilise the equilibria of complex ecosystems by shrinking part of the bulk region of the eigenvalue spectrum. The main result of our work is that the combination of trophic structure and dispersal can destabilise such equilibria via the outlier eigenvalues. The Referee’s question is, in essence, if and how these two sets of results can be reconciled with each other. We note that GML’s model does not include trophic structure (designated groups of predator and prey species), while we had initially not considered spatial heterogeneity.

In order to answer this question, we have followed Dr Barabás’ suggestion and have constructed a model which combines the features of our setup with that of GML, i.e., dispersal, trophic structure and spatial heterogeneity. A detailed description can be found in Section S10 A of the revised Supplementary Material. To analyse this model, we extended the analytical random-matrix approach used to calculate the eigenvalue spectrum. The resulting equations for the bulk and outlier eigenvalues in Sections S10 B and S10 C then provide the starting point for a more comprehensive analysis.

This analysis allows us to correct an aspect of the Discussion in our original submission. We had initially reported an apparent contrast between the results of our work and those of GML. As Dr Barabás suspects in his report these two sets of results are not incompatible with each other. Our initial attribution of this contrast to the differences in the implementation of dispersal is hence not justified, and we have amended the Discussion accordingly. More precisely, we reproduce the results of GML from the extended model using our analytical approach (Section S11 B of the revised Supplement). We also show that the conclusions of GML carry over to the case where hopping is local as opposed to all-to-all (see Section S12 A). Conversely we show in Section S13 that the destabilising mechanism continues to be possible when dispersal is implemented as in GML (all-to all).

From the additional analyses we see that both dispersal-induced stabilisation (as described by GML) and dispersal-induced instability are possible and can coexist within the same model. We demonstrate this in Figures S13 and S14 of the revised Supplement. Precisely which effect is observed is dependent on the model parameters. The contributing factors to determining whether dispersal is a stabilising or destabilising influence are discussed in Section S14 of the Supplement. In short, the stabilising effects of dispersal are promoted by a large typical dispersal rate and high levels of spatial heterogeneity in the interaction coefficients. The destabilising effects of dispersal are promoted by trophic structure and a significant predator-to-prey ratio of dispersal rates. This is illustrated in Fig. S15 of the revised Supplement.

In summary, we have now developed a broader picture of the matter, and we would like to thank Dr Barabás for prompting us to do this. Gravel et al have – quite rightly – reported dispersal as a factor which can stabilise equilibria of complex ecosystems in the right circumstances. This is in-line with work aiming to restore stability in May’s model through the inclusion of further aspects of natural ecosystems. The purpose of our work is to highlight that dispersal can also act to destabilise equilibria of complex ecosystems with trophic structure.

Before we turn to the more specific items in Dr Barabás’ comments 3(a)–(c), we’d like to stress that it was not our intention to suggest that the work by GML was inaccurate in any way. In fact, we confirm the results for the eigenvalue spectra and the stability criteria presented in GML with our method in Equations (S177) – (S179) and (S187) respectively.

Specific response to 3(a):

We have done what the Referee proposes and have reconciled the two models. In specific response to this comment, we have constructed a chain of patches with hopping between adjacent patches and with heterogeneity across the interaction matrices in different patches. We have extended the analytical formalism to cover this scenario (it is a special case of the more general results derived in Section S10 of the revised Supplement). We do indeed find that the results of GML carry over to the case where hopping is implemented in a ‘nearest neighbour’ fashion as opposed to ‘all-to-all’. This is shown in Section S12 A in the Supplement.

As the Referee suspects and as is described above in our general response to 3(a)–(c), the conclusions by GML and ours are compatible with each other and, indeed, the precise implementation of hopping is largely immaterial. Instead, the two models describe two complementary situations.

Specific response to 3(b):

Although the Referee is correct that a discrete patch landscape does restrict the range of wavelengths that can propagate, restrictions on the allowed wave numbers are not the primary reason for the absence of dispersal-induced instability in the model by GML. This can be seen from Section S13, where we show that dispersal-induced instability persists in models defined on discrete patches. Instead, the reason is that the model in Ref. [R1] does not allow for trophic structure and outlier eigenvalues are not considered in this work.

To answer the second part of the Referee’s question we stress that we intentionally use a simple stylised model to highlight the mechanism leading to dispersal-induced instability. This parsimonious approach is very much in line for example with [R1, R2, R8, R9], and indeed with May [R10]. It sacrifices some amount of realism, but improves theoretical clarity. With this in mind, the model parameters are dimensionless, and the purpose is not to capture the aspects of any particular real ecosystem quantitatively. At the same time, we note that spatial instabilities and patterns commensurate with the Turing mechanism have been observed in a number of natural ecosystems at different length scales (centimetres to tens of meters in arid and marine systems, see e.g. [R11–R15], smaller scales in communities of bacteria [R16]). This demonstrates that Turing-like instabilities can occur in real-world ecosystems at meaningful wavelengths. It is then reasonable to assume that the potential for dispersal-induced instability carries over to complex natural ecosystems with more species. We have added a brief comment to this effect in the second paragraph of the Discussion.

Specific response to 3(c):

As the Referee anticipated, the two approaches are compatible. Our additional analysis shows that

the seemingly different outcomes are not due the implementation of dispersal. As a consequence, we have removed the comments to this effect in the original Discussions section. Instead, we now give a more inclusive account of the effects of dispersal on stability and describe how our results fit in with those presented by GML [R1].

Changes made to the manuscript in response to 3(a)–(c):

In response to the comments we have added Sections S10-S14 to the Supplement, detailing how spatial heterogeneity can be added to our model and how our analytical approach can be extended to cover this more general case. We also describe in more depth how our work fits in with that of GML [R1]. We have modified the manuscript accordingly. Particularly we have rephrased the Discussion. We have added the paragraph titled ‘Inclusion of spatial heterogeneity in the interaction coefficients’ in Methods.

(4) Lastly, the way the trophic levels are implemented is somewhat suspect. The problem is not with having two strict levels. Rather, it seems that feeding interactions are present even within a single level, since the interaction matrices within a level are generated as random matrices drawn from Gaussian ensembles. In principle therefore, nothing prevents two prey species having a (+,-) relationship with one another (intraguild predation). Is this indeed the case? If so then I would either think more about the interpretation of the blocks (which are not necessarily trophic levels sensu stricto), or else restrict entries within levels to exclude trophic interactions.

Response:

Again, we agree with Dr Barabás. The different blocks do not necessarily correspond to strict trophic levels. In addition to possible intraguild predation not all reactions between species in the predator class with a species in the prey class are necessarily of type (+,-). This is an inevitable consequence of the choice of an underlying bi-variate Gaussian distribution, whose support stretches across all combinations of real values. Instead, the two groups of species represent a statistical structure in the population in the following sense. The interactions between a ‘predator’ species with a ‘prey’ species have a statistical tendency to be trophic, but this need not be the case in all instances. We note that our setup is not too dissimilar to that used to describe communities of animals and plants in Grilli et al [R17] [see in particular their Eq. (13) and the subsequent text].

With this being said, we now show explicitly in the revised Supplemental Material (Section S8) that it is possible to strictly enforce a rigid trophic structure, ensuring a given combination of signs for pairs of elements in the random matrix. Related to Dr Barabás’ point (5) below, this is made possible by the universality of our theory (see revised Supplement Section S7). We show that our predictions for the eigenvalue spectra carry over to the case where predator and prey species are required to have a (+,-) relationship with one another and where no such predator-prey pairings are possible within either group (i.e., intraguild predation is eliminated). We believe this lends credence to our interpretation of the blocks as relating to trophic structure.

Changes made to the manuscript:

We have softened the language, and no longer speak of ‘trophic levels’. As there Referee says, this could be understood to mean strictly trophic interactions in all cases. Instead we think that ‘trophic

structure' is a more appropriate description, and this is what we use in the revised manuscript. This point is discussed further in the new Section S8 in the Supplemental Material, where we show how the analytical formalism can be applied to models with given combinations of signs and also show that dispersal-induced instability continues to be possible.

(5) About the methodology, I wanted to bring up two points. The first concerns universality. While the random parts of the matrices are always drawn from Gaussian distributions, it is implied at several places that the results are universal (i.e., under mild conditions, only the mean, variance, and correlation structure of the distribution matters, not its precise shape). Is that really so? Universality has been proven for circular and elliptic matrices, but these are difficult proofs and, to my knowledge, have not yet been generalized to more complicated random matrices. Is universality a theorem here, or more of a numerically supported and highly reasonable conjecture? I would state which it is either way. Also, if it is the latter, then I would do a few quick numerical studies, drawing matrix entries from two different distributions with the same means, variances, and covariances, to verify that they lead to the same asymptotic eigenvalue distributions.

Response:

We thank Dr Barabás for this suggestion. While we think our original manuscript made only peripheral reference to universality, we certainly had this in mind tacitly as a sensible conjecture. The Referee's comments have prompted us to investigate this in more detail. We do indeed find that results derived for the case of Gaussian randomness carry over to more general cases, provided suitable (mild) conditions on the moments of the distribution are fulfilled.

While rigorously proving universality in the sense of mathematics say to the degree of Ref. [R18] is not within the scope of our work, we present what we believe to be a convincing analytical argument in Section S7 of the revised Supplementary Material. Numerical evidence is shown in the new Figure S2 of the Supplement, where we have drawn the interaction matrix elements from Bernoulli, Laplace and uniform distributions.

Changes made to the manuscript:

A discussion of universality, our analytical argument, and numerical verification can be found in Section S7 of the revised Supplementary Material. We mention universality more clearly in the revised main text, particularly in the Discussion and also where the model is introduced [near Eq. (2)].

(6) The second point is about how the method developed by the Authors connects to existing ones. In my mind, the methodological contribution is as important (if not more so) than the biological one, since it enables one to determine the spectral density and spectral abscissa of a new class of matrices. But then, it would help to carefully point out how and where the method differs from, and fits in with, seemingly very similar contributions. For instance, Allesina et al. (2015 Nat Commun) and Grilli et al. (2017 Nat Commun) discuss in depth how to evaluate the spectral density functions of matrices with block structure (corresponding to the two trophic levels here), and Barabás et al. (2017 Nature Ecol Evol) discuss the resolvent method and how to take unequal diagonal entries into account. While the Supplement is didactically very well written, it might be worth putting more emphasis on what the methodological contribution of this manuscript is, in relation to its predecessors. This would

make it easier for people to know under what circumstances its results can and cannot be used, and whether multiple different approaches could be used to reach the same conclusions.

Response:

We share the Referee’s opinion on the importance of methodological advances. We also agree that it is useful to provide a little more context to our statistical-mechanics-inspired approach to calculating the eigenvalue support, and to state more clearly what the novelty is on the technical level.

The method we use builds on the work by Sommers et al [R19] who map the problem of finding the resolvent of the random matrices onto a high-dimensional integral which is solved using the saddle-point method (see e.g. [R20–R22] for modern accounts and applications). The main methodological novelty of our work is the introduction of the block structure in the matrix. This is dealt with conveniently through the introduction of integration variables with an additional block index α . In principle, the approach can be used for matrices with more complicated structure. We also show how quantities from the calculation of the bulk spectrum can be repurposed to find the outlier eigenvalues.

The approach has further advantages. For example, the origin of universality is transparent (Section S7 A of the revised Supplement). The strength of the approach is further demonstrated by the ability to reproduce and extend results in Refs. [R2] and [R1], obtained by two different methods. Details can be found in Sections S9, S11 B and S12 in the Supplementary Information.

We now briefly discuss each of the works mentioned by the Referee:

(i) The work by Allesina et al. (2015 Nat Commun) [R9] evaluates the eigenvalue spectra of random matrices with cascade structure. The elements above the diagonal have different mean and variance to the elements below the diagonal. The matrices in our model instead are divided into four rectangular blocks with differing statistics, rather than two triangles, so to speak. The analysis in Ref. [R9], while successful, relies on assumptions which break down in certain limits of the system parameters. For example, it is assumed that Weyl’s inequality for symmetric matrices carries over to this case and that the bulk eigenvalues can be approximated by an ellipse. We make no assumptions other than to assume that the number of species N is large ($N \gg 1$). We think that the application of the techniques used in our work to matrices like those in Ref. [R9] is an intriguing prospect for future work.

(ii) The work by Grilli et al [R17] involves matrices with similar block structure to those in our model. The blocks are associated with plant and animal species in [R17], similar to the prey and predator structure in our manuscript. Grilli et al employ methods from statistical mechanics but, instead of evaluating eigenvalue spectra, they compute the number of growth vectors leading to feasible equilibria. We also note that there is no spatial interaction in the model in [R17].

As an aside, we remark that the calculation in [R17] is in a similar spirit to the work by Elizabeth Gardner [R23] on the storage capacity of random neural networks. (Statistical physicists would refer to the calculation by Grilli et al as a ‘Gardner calculation’). We add that one of us has worked on separate (but related) calculations of the number of Nash equilibria in random games [R24]. Estimating the fraction of growth vectors leading to feasible equilibria, or indeed calculating directly the number of (stable) feasible equilibria in complex ecosystems with or without spatial dispersal, is again an interesting line for future research.

(iii) The work [R2] uses the method of quaternionic resolvents primarily to deduce the spectral abscissa of elliptic random matrices with randomly drawn diagonal elements. The full boundary of the eigenvalue support is also readily available from [R2]. As mentioned in the response to comment (2) by Referee 1 we use our method to reproduce and extended these results in Section S9 of the revised Supplement.

We imagine that the method of quaternionic resolvents could also be used to derive the main results in our work, although we have not attempted to explore how exactly the block structure can be incorporated. Matrices with more complex internal structure appear to be an interesting opportunity to further develop the quaternionic approach.

Changes made to the manuscript:

We have added a new subsection to the Supplement (Section S2 A) to provide more context to our approach. In this section we also describe what the methodological advances are in relation to existing work on the calculation of spectra of random matrices, in particular with methods from statistical physics and the theory of disordered systems.

The three papers mentioned by the Referee are now highlighted in the main manuscript and in the revised Supplement. The works by Alessina et al (2015) and by Grilli et al (2017) are mentioned in the main paper (in the text before Eq. (2)). We have also added a paragraph at the end of Section S1 A in the Supplement to briefly discuss the relation of our work to these existing references. In the revised submission we have also carried out an analysis of a model with variation in the diagonal elements (this can be found in the new Section S9 in the Supplementary Material). We mention this just before Eq. (3) in the main paper and in ‘Further variations of the model’, and we refer to the paper by Barabás et al (2017).

(7) l.21-23: I would try to avoid falling in the trap of giving a naïve account of the supposed naïveté of the cited works. Such overwhelming confidence about one’s own intuition is definitely absent from e.g. MacArthur (1955 Ecology). I suggest starting the paper with "Providing a firm counterpoint to the view that greater ecosystem complexity promoted their stability [1-6], Robert May used ..." (or something to that effect).

Response and changes made:

We agree with the Referee’s comment and thank him for bringing this to our attention. We have modified the opening paragraph accordingly.

(8) l.56-57: "variation in interaction strength" - that was surely one thing that May’s original model did capture, or am I missing something?

Response and changes made:

Yes, we agree. This was unfortunate wording on our part – we meant to say ‘alternative interpretations of interaction strength’. We thank the Reviewer for spotting this. We have amended the text.

(9) Fig. 2 caption: I would write "rightmost eigenvalue" instead of "most unstable eigenvalue".

Response and changes made:

Yes, we agree, thanks for bringing this to our attention. We have changed this.

(10) l.250: *like* → *likely*

Response and changes made:

Done. Thank you for spotting this!

(11) l.272: *This is not actually a necessary condition for stability. When the correlation between the (i,j) th and (j,i) th entries of the matrix is strongly negative, then it is possible for at least some diagonal entries to be positive and for the system to remain stable (see Barabás et al. 2017 Nature Ecol Evol for details).*

Response and changes made:

Indeed, as pointed out by Barabás et al [R2], this is not strictly necessary for stability. We thank the Referee for bringing this to our attention. We have removed the misleading sentence.

III. FURTHER CHANGES

We have made small changes throughout, for example to correct language where necessary, or to make the presentation more clear.

-
- [R1] Dominique Gravel, François Massol, and Mathew A Leibold, “Stability and complexity in model meta-ecosystems,” *Nature Communications* **7**, 12457 (2016).
 - [R2] György Barabás, Matthew J Michalska-Smith, and Stefano Allesina, “Self-regulation and the stability of large ecological networks,” *Nature Ecology & Evolution* **1**, 1870–1875 (2017).
 - [R3] Ilkka Hanski, “Metapopulation dynamics,” *Nature* **396**, 41–49 (1998).
 - [R4] Ilkka Hanski, “Habitat connectivity, habitat continuity, and metapopulations in dynamic landscapes,” *Oikos* **87**, 209–219 (1999).
 - [R5] Shai Pilosof, Mason A Porter, Mercedes Pascual, and Sonia Kéfi, “The multilayer nature of ecological networks,” *Nature Ecology & Evolution* **1**, 1–9 (2017).
 - [R6] Matthew C Hutchinson, Bernat Bramon Mora, Shai Pilosof, Allison K Barner, Sonia Kéfi, Elisa Thébault, Pedro Jordano, and Daniel B Stouffer, “Seeing the forest for the trees: Putting multilayer networks to work for community ecology,” *Functional Ecology* **33**, 206–217 (2019).
 - [R7] Alan M Turing, “The chemical basis of morphogenesis,” *Philosophical Transactions of the Royal Society of London. Series B, Biological Sciences* **237**, 37–72 (1952).
 - [R8] Jacopo Grilli, Tim Rogers, and Stefano Allesina, “Modularity and stability in ecological communities,” *Nature Communications* **7**, 1–10 (2016).
 - [R9] Stefano Allesina, Jacopo Grilli, György Barabás, Si Tang, Johnatan Aljadeff, and Amos Maritan, “Predicting the stability of large structured food webs,” *Nature Communications* **6**, 1–6 (2015).
 - [R10] Robert M May, “Will a large complex system be stable?” *Nature* **238**, 413–414 (1972).
 - [R11] Max Rietkerk, Stefan C. Dekker, Peter C. de Ruiter, and Johan van de Koppel, “Self-organized patchiness and catastrophic shifts in ecosystems,” *Science* **305**, 1926–1929 (2004).
 - [R12] Max Rietkerk and Johan van de Koppel, “Regular pattern formation in real ecosystems,” *Trends in Ecology and Evolution* **23**, 169–175 (2008).

- [R13] Johan van de Koppel, Joanna C. Gascoigne, Guy Theraulaz, Max Rietkerk, Wolf M. Mooij, and Peter M. J. Herman, “Experimental evidence for spatial self-organization and its emergent effects in mussel bed ecosystems,” *Science* **322**, 739–742 (2008).
- [R14] Ehud Meron, “Pattern-formation approach to modelling spatially extended ecosystems,” *Ecological Modelling* **234**, 70 – 82 (2012).
- [R15] Q. Liu, P. Herman, W. Mooij, *et al.*, “Pattern formation at multiple spatial scales drives the resilience of mussel bed ecosystems,” *Nature Communications* **5**, 5234 (2014).
- [R16] David Karig, K. Michael Martini, Ting Lu, Nicholas A. DeLateur, Nigel Goldenfeld, and Ron Weiss, “Stochastic turing patterns in a synthetic bacterial population,” *Proceedings of the National Academy of Sciences* **115**, 6572–6577 (2018).
- [R17] Jacopo Grilli, Matteo Adorisio, Samir Suweis, György Barabás, Jayanth R Banavar, Stefano Allesina, and Amos Maritan, “Feasibility and coexistence of large ecological communities,” *Nature Communications* **8**, 14389 (2017).
- [R18] Terence Tao, Van Vu, Manjunath Krishnapur, *et al.*, “Random matrices: Universality of ESDs and the circular law,” *The Annals of Probability* **38**, 2023–2065 (2010).
- [R19] Hans Juergen Sommers, Andrea Crisanti, Haim Sompolinsky, and Yaakov Stein, “Spectrum of large random asymmetric matrices,” *Physical Review Letters* **60**, 1895–1898 (1988).
- [R20] Alexander Altland and Ben D Simons, *Condensed Matter Field Theory* (Cambridge University Press, Cambridge, UK, 2010).
- [R21] Marc Mézard, Giorgio Parisi, and Miguel Virasoro, *Spin glass theory and beyond: An Introduction to the Replica Method and Its Applications*, Vol. 9 (World Scientific Publishing Company, London, 1987).
- [R22] Antonius C. C. Coolen, *The Mathematical Theory of Minority Games: Statistical Mechanics of Interacting Agents* (Oxford University Press, Oxford, UK, 2005).
- [R23] Elizabeth Gardner, “The space of interactions in neural network models,” *Journal of physics A: Mathematical and General* **21**, 257 (1988).
- [R24] Tobias Galla, “Two-population replicator dynamics and number of nash equilibria in matrix games,” *EPL (Europhysics Letters)* **78**, 20005 (2007).

Reviewers' Comments:

Reviewer #2:

Remarks to the Author:

The Authors have done an excellent, very careful job of addressing all previous comments. The manuscript still reads as well as its predecessor, but the results are much strengthened and clarified. I am very happy to recommend it for publication.

Sincerely,
Gyuri Barabás